# Characterization of PM$_{10}$ Emission Rates from Roadways in a Metropolitan Area Using the SCAMPER Mobile Monitoring Approach

**Dennis R. Fitz * and Kurt Bumiller**

Center for Environmental Research and Technology, College of Engineering, University of California, 1084 Columbia Avenue, Riverside, CA 92507, USA; kurt@kurtbumiller.com
*   Correspondence: dfitz@cert.ucr.edu

**Abstract:** The SCAMPER mobile system for measuring PM$_{10}$ emission rates from paved roads was used to characterize emission rates from a wide variety of roads in the Phoenix, AZ metropolitan area. Week-long sampling episodes were conducted in March, June, September, and December. A 180 km-long route was utilized and traveled a total of 18 times. PM$_{10}$ emission rate measurements were made at 5-s resolution for over 3200 km of roads with a precision of approximately 25%. The PM$_{10}$ emission rates varied by over two orders of magnitude and were generally low unless the road was impacted with dust deposited by activities such as construction, sand and gravel operations, agriculture, and vehicles traveling on or near unpaved shoulders and roads. The data were tabulated into averages for each of 67 segments that the route was divided into. The segment-averaged PM$_{10}$ emission rates ranged from zero to 2 mg m$^{-1}$, with an average of 0.079 mg m$^{-1}$. There was no significant difference in emission rates between seasons. There was a major drop in emission rates over a weekend, when dust generation activities such as construction are expected to be much reduced. By Monday, the PM$_{10}$ emission rates had risen to the levels of the previous Friday. This indicates that roads quickly reach an equilibrium PM$_{10}$ generating potential.

**Keywords:** PM$_{10}$; road dust; fugitive dust; particulate matter; paved roads; emission rates

## 1. Introduction

Particulate matter (PM) has been shown by epidemiological studies to be responsible for premature deaths [1]. The U.S. Environmental Protection Agency has set air quality standards for particles both of less than 10 μm and 2.5 μm in aerodynamic diameter, PM$_{10}$ and PM$_{2.5}$, respectively. Many government agencies have adopted these standards or have derived similar ones. Many of these standards are exceeded in urban areas and effective mitigation methods are necessary to meet these standards. In order to implement cost-effective control strategies, the sources of the PM must be determined as accurately as possible. Models have estimated that a significant amount of this material can originate from paved roadways [2–4].

Measurement of emission rates from fugitive sources such as PM from vehicles on roadways cannot be measured directly, but must be estimated. This has been done using dispersion modeling [5–7], receptor modeling [8] a combination of dispersion and receptor modeling [9,10], tracer studies [11–13], and measuring the flux of PM$_{10}$ through a horizontal plane downwind of the source [14–17]. All of these methods require significant resources to characterize the emissions from actual roadways for inventory development in addition to presenting large uncertainties in the results.

The U.S. Environmental Protection Agency (EPA) used detailed measurements of PM flux through a plane for estimating the PM emissions from paved roads to derive an empirical equation using surface silt loading and vehicle weight as metrics [18]. This equation contains significant amounts of uncertainty and the EPA has revised it several

times over the past decades based on reviews of the methods used. The current equation is as follows:

$$E = k(sL/)^{0.91} (W)^{1.02}$$ (1)

where:

E = Particulate matter emission rate in the units of g/VKT
k = A constant dependent on the aerodynamic size range of PM (0.62 for $PM_{10}$)
sL = Road surface silt loading of material smaller than 75 μm in g m$^{-2}$
W = mean vehicle weight in U.S. tons
VKT = vehicle kilometer traveled

Despite the uncertainties in this equation, it is widely used to estimate emission inventories in air basins. Compounding this, since the determination of silt loading is labor intensive and often dangerous, EPA default values for silt loading are often used to estimate emission rates for emission inventories. More recently, the direct measurement of PM emissions in real time using a moving vehicle have been reported. In one version, TRAKER™, the concentration of $PM_{10}$ is measured in the wheel well of a moving vehicle [19]. This value is then related to an emission rate by calibrating with a downwind measurement of $PM_{10}$ flux. Other investigators have used the TRAKER approach specialized for their own studies [20,21]. In another approach, SCAMPER (System for the Continuous Aerosol Measurement of Particulate Emissions from Roadways), the $PM_{10}$ concentration is measured in front of a vehicle and a representative point in the wake behind it [22]. With this approach, as a first approximation the emission rate is determined by multiplying the net concentration difference by the frontal area of the vehicle. Both TRAKER and SCAMPER have been evaluated together using a dedicated roadway on which known amounts of soil were evenly deposited [23]. Measurements of $PM_{10}$ emission rates were concurrently made using the flux method and measuring the silt loading. Reasonable agreement was found between all of these methods.

The objective of this study was to use the SCAMPER to:

1. Provide actual measurements of $PM_{10}$ emission rates from roadways that could be used to construct a data-based emission inventory.
2. Evaluate the significance of construction activities on $PM_{10}$ emission rates.
3. Determine if there are seasonal changes in the emission rates.
4. Evaluate the precision of the measured emission rates.

## 2. Materials and Methods

### 2.1. Test Route

The testing was conducted over streets in the greater Phoenix, AZ metropolitan area. This climate is typical of the Sonoran Desert with less than 13 cm of precipitation per year. No rain or other significant weather events occurred before or during the test periods. Except as noted, wind speeds were generally less than 10 km h$^{-1}$, similar to the conditions when the SCAMPER was operated under controlled conditions [23]. Temperatures, which were not expected to affect SCAMPER measurements, ranged from 13 °C in the winter to 33 °C in the summer. The route consisted of a mix of segments of different road types based on their Average Daily Traffic (ADT) or number of vehicles in both directions passing a point per day. Most segments were at least a half kilometer long so that time-integrated measurements could be collected with reasonable uncertainty. Five segment types were differentiated:

I: Less than 10,000 ADT: 43 km total
II: 10,000–19,999 ADT: 48 km total
III: 20,000–29,000 ADT: 12 km total
IV: Greater than 30,000 ADT: 7 km total
Limited Access: 70 km total

The route included representative lengths of all road classes (I–IV) and the limited access or freeway (Fwy). The total length was 180 km. Figure 1 is a map of the test route

and Table 1 identifies each segment, what class of road it belongs to, and what type of land use area it was located in. The SCAMPER was driven at a speed corresponding to the general flow of traffic.

### 2.2. SCAMPER Description

The SCAMPER determines PM emission rates from roads by measuring the PM concentrations in front of (mounted on the hood) and behind the vehicle (mounted on a small open trailer) using optical sensors with a 1s time resolution. The system and its validation have previously been described [22,23]. Briefly, the SCAMPER, shown in Figure 2, includes five major components:

Tow vehicle and Trailer: A 1994 Chevrolet Suburban was used to tow a small (3.1 m wide by 2 m long) open flatbed trailer. The trailer was fitted with a 1 m hitch extension to place the rear sampling inlet 3 m behind the tow vehicle at a height of 0.8 m above the ground on the centerline of the trailer. This position was found to give $PM_{10}$ concentrations that were representative of the mean concentration of $PM_{10}$ in the wake of the tow vehicle [22].

$PM_{10}$ Sensors: Thermo Systems Inc. (Shoreville, MN, USA) Model 8520 DustTrak™ optical PM sensors with $PM_{10}$ inlets.

Isokinetic Sampling Inlets: A custom made inlet where the inlet speed is matched to the air speed by a laptop computer that monitors the static air pressure and adjusts the inlet pressure to match it by controlling a vacuum pump (mounted on the trailer). This condition creates a no-pressure-drop inlet; therefore, the sampled air stream has the same energy as the ambient air stream.

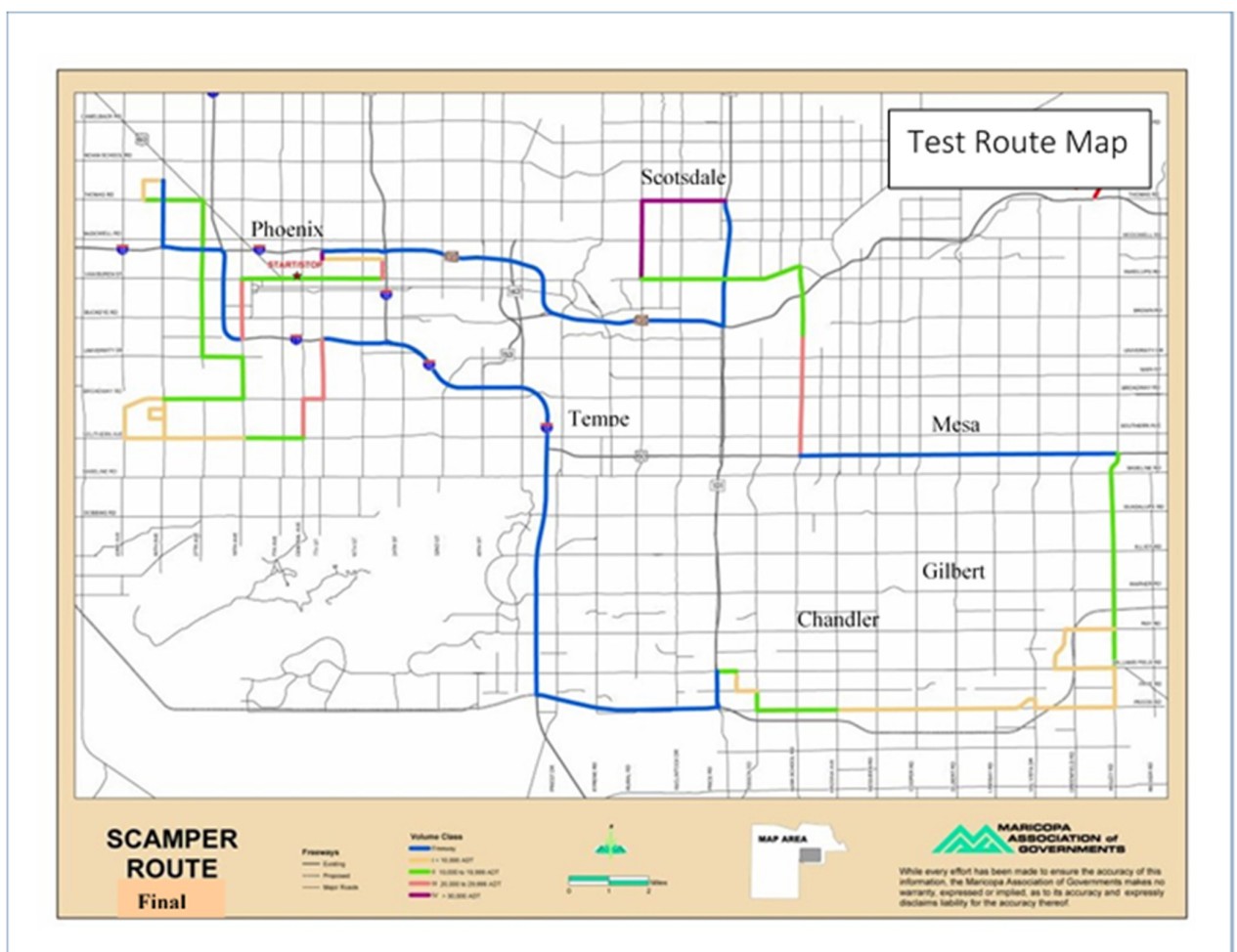

**Figure 1.** Map of the test route.

**Table 1.** List of test route segments and road classification type.

| Seg # | Intersection | On Street | Dir | Length km | From Street | To Street | Vol Class | # of Lanes | Predominant Land Use |
|---|---|---|---|---|---|---|---|---|---|
| | Begin: 1st Ave/Van Buren | 1st Ave | SB | — | | | II | 3 | Commercial |
| 1 | 1st Ave/Van Buren | Van Buren St | EB | 3.2 | 1st Ave | 20th St | II | 2 | Commercial |
| 2 | Van Buren/20th St | 20th St | NB | 0.8 | Van Buren St | Roosevelt St | III | 5 | Mixed |
| 3 | 20th St/Roosevelt St | Roosevelt St | WB | 2.4 | 20th St | 7th St | I | 1 | Residential |
| 4 | Roosevelt/7th St | 7th St | NB | 0.8 | Roosevelt | I-10E on-ramp | IV | 3 | Mixed |
| 5 | I-10 East/7th St | I-10 East | EB | 1.6 | 7th St | SR 202 | Fwy | | Mixed |
| 6 | I-10 East/SR 202 | SR 202 | EB | 14.4 | SR 202 | SR 101 | Fwy | | Mixed |
| 7 | SR 202/SR 101 | SR 101 | NB | 2.4 | SR 202 | Thomas Rd | Fwy | | Agricultural |
| 8 | SR 101/Thomas Rd | Thomas Rd | WB | 3.2 | SR 101 | Scottsdale Rd | IV | 2 | Residential |
| 9 | Thomas/Scottsdale Rd | Scottsdale Rd | SB | 3.2 | Thomas Rd | McKellips Rd | IV | 3 | Comm/Res |
| 10 | Scottsdale/McKellips Rd | McKellips Rd | EB | 6.7 | Scottsdale Rd | Alma School Rd | II | 2 | Agricultural |
| 11 | McKellips/Alma School | Alma School Rd | SB | 2.9 | McKellips Rd | 8th St | II | 3 | Industrial |
| 12 | | Alma School Rd | SB | 4.8 | 8th St | US 60 | III | 3 | Commercial |
| 13 | Alma School/US 60 | US 60 | EB | 12.6 | Alma School Rd | Higley Rd | Fwy | | Mixed |
| 14 | US 60/Higley | Higley Rd | SB | 2.4 | US 60 | Guadalupe Rd | II | 3 | Agricultural/Res |
| 15 | | Higley Rd | SB | 1.6 | Guadalupe Rd | Elliot Rd | II | 3 | Agricultural/Res |
| 16 | | Higley Rd | SB | 1.6 | Elliot Rd | Warner Rd | II | 1 | Agricultural/Res |
| 17 | | Higley Rd | SB | 1.6 | Warner Rd | Ray Rd | II | 1 | Agricultural/Res |
| 18 | | Higley Rd | SB | 1.6 | Ray Rd | Williams Field Rd | II | 1 | Agricultural/Res |
| | **Inner Loop #1** | | | | | | | | |
| 19 | Higley/Williams Field Rd | Williams Field Rd | WB | 2.4 | Higley Rd | Santan Valley Pky | I | 1 | Agricultural/Res |
| 20 | Williams Field/Santan Valley | Santan Valley Pky | NB | 1.8 | Williams Field Rd | Ray Rd | I | 1 | Agricultural/Res |
| 21 | Santan Valley/Ray Rd | Ray Rd | EB | 2.4 | Santan Valley Pky | Higley Rd | I | 1 | Agricultural/Res |
| 18 | Ray Rd/Higley Rd | Highley Rd | SB | 1.6 | Ray Rd | Williams Field Rd | I | 1 | Agricultural/Res |
| 22 | | Higley Rd | SB | 1.6 | Williams Field Rd | Pecos Rd | I | 1 | Agricultural/Res |
| 23 | Higley/Pecos Rd | Pecos Rd | WB | 1.6 | Higley Rd | Greenfield Rd | I | 3 | Agricultural/Res |
| 24 | | Pecos Rd | WB | 1.8 | Greenfield Rd | Val Vista Rd | I | 3 | Agricultural/Res |
| 25 | | Pecos Rd | WB | 1.8 | Val Vista Rd | Lindsay Rd | I | 2 | Agricultural/Res |
| 26 | | Pecos Rd | WB | 1.4 | Lindsay Rd | Gilbert Rd | I | 1 | Agricultural/Res |
| 27 | | Pecos Rd | WB | 1.6 | Gilbert Rd | Cooper Rd | I | 2 | Agricultural/Res |
| 28 | | Pecos Rd | WB | 1.6 | Cooper Rd | McQueen Rd | I | 1 | Agricultural/Res |
| 29 | | Pecos Rd | WB | 1.6 | McQueen Rd | Arizona Ave | I | 2 | Agricultural/Res |
| 30 | | Pecos Rd | WB | 1.6 | Arizona Ave | Alma School Rd | II | 1 | Agricultural/Res |
| 31 | | Pecos Rd | WB | 1.6 | Alma School Rd | Dobson Rd | II | 1 | Agricultural/Res |
| 32 | Pecos Rd/Dobson | Dobson Rd | NB | 0.6 | Pecos Road | Frye Rd | II | 1 | Commercial |
| 33 | Dobson Rd/Frye Rd | Frye Rd | WB | 0.8 | Dobson Rd | Ellis Rd | I | 1 | Commercial |
| 34 | Frye Rd/Ellis Rd | Ellis Rd | NB | 0.6 | Frye Rd | Chandler Blvd | I | 1 | Commercial |
| 35 | Ellis/Chandler Blvd | Chandler Blvd | WB | 0.8 | Ellis Rd | Price Freeway | II | 3 | Commercial |
| 36 | Chandler Blvd/Price Fwy | Price Frontage Rd | SB | 1.4 | Chandler Blvd | Santan Freeway | Fwy | | Commercial |
| 37 | Price Fwy/Santan Fwy | Santan Freeway | EB | 8.2 | Price Freeway | I-10 West | Fwy | | Mixed |
| 38 | Santan Fwy/I-10 West | I-10 West | NB | 17.6 | Santan Freeway | I-17 West | Fwy | | Mixed |
| 39 | I-10 West/I-17 West | I-17 West | WB | 3.2 | I-17W interchange | 7th St off-ramp | Fwy | | Mixed |
| 40 | I-17 West/7th Street | 7th St | SB | 2.4 | 7th St off-ramp | Broadway Rd | III | 2 | Mixed |
| 41 | 7th St/Broadway Rd | Broadway Rd | WB | 0.8 | 7th St | Central Ave | III | 2 | Mixed |
| 42 | Broadway/Central Ave | Central Ave | SB | 1.6 | Broadway Rd | Southern Ave | III | 2 | Mixed |
| 43 | Central/Southern Ave | Southern Ave | WB | 2.4 | Central Ave | 19th Ave | II | 1 | Residential |
| 44 | | Southern Ave | WB | 1.6 | 19th Ave | 27th Ave | I | 1 | Residential |
| 45 | | Southern Ave | WB | 1.6 | 27th Ave | 35th Ave | I | 1 | Residential |
| 46 | | Southern Ave | WB | 1.6 | 35th Ave | 43rd Ave | I | 1 | Industrial |
| 47 | Southern/43rd Ave | 43rd Ave | NB | 1.3 | Southern Ave | Broadway Rd | I | 1 | Industrial |
| 48 | 43rd Ave/Broadway Rd | Broadway Rd | EB | 1.9 | 43rd Ave | 35th Ave | I | 1 | Industrial |
| | **Inner Loop #2** | | | | | | | | |
| 49 | Broadway/35th Ave | 35th Ave | SB | 1.6 | Broadway Rd | Southern Ave | I | 1 | Industrial |
| 50 | 35th Ave/Southern Ave | Southern Ave | WB | 1.6 | 35th Ave | 43rd Ave | I | 1 | Industrial |
| 47 | Southern/43rd Ave | 43rd Ave | NB | 1.3 | Southern Ave | Broadway Rd | I | 1 | Industrial |
| 48 | 43rd Ave/Broadway Rd | Broadway Rd | EB | 1.9 | 43rd Ave | 35th Ave | I | 1 | Industrial |
| 51 | Broadway/35th Ave | 35th Ave | SB | 0.5 | Broadway Rd | Wier Ave | II | 1 | Commercial |
| 52 | 35th Ave/Wier Ave | Wier Ave | WB | 0.6 | 35th Ave | 38th Ave | I | 1 | Residential |
| 53 | Wier Ave/38th Ave | 38th Ave | SB | 0.5 | Wier Ave | Roeser Rd | I | 1 | Residential |
| 54 | 38th Ave/Roeser Rd | Roeser Rd | EB | 0.6 | 38th Ave | 35th Ave | I | 1 | Residential |
| 55 | Roeser/35th Ave | 35th Ave | NB | 0.8 | Roeser Rd | Broadway Rd | II | 1 | Agricultural/Res |
| 56 | 35th Ave/Broadway | Broadway Rd | EB | 3.2 | 35th Ave | 19th Ave | II | 1 | Industrial |
| 57 | Broadway/19th Ave | 19th Ave | NB | 1.6 | Broadway Rd | Lower Buckeye Rd | II | 1 | Industrial |
| 58 | 19th Ave/Lower Buckeye | Lower Buckeye Rd | WB | 1.6 | 19th Ave | 27th Ave | II | 1 | Industrial |
| 59 | Lower Buckeye/27th Ave | 27th Ave | NB | 1.6 | Lower Buckeye Rd | Buckeye Rd | II | 2 | Industrial |
| 60 | | 27th Ave | NB | 1.6 | Buckeye Rd | Van Buren St | II | 2 | Industrial |
| 61 | | 27th Ave | NB | 1.6 | Van Buren St | McDowell Rd | II | 2 | Industrial |
| 62 | | 27th Ave | NB | 1.6 | McDowell Rd | Thomas Rd | II | 2 | Industrial |
| 63 | 27th Ave/Thomas Rd | Thomas Rd | WB | 2.4 | 27th Ave | 39th Ave | II | 3 | Commercial |
| 64 | Thomas/39th Ave | 39th Ave | NB | 0.8 | Thomas Rd | Osborn Rd | I | 1 | Residential |
| 65 | 39th Ave/Osborn Rd | Osborn Rd | EB | 0.8 | 39th Ave | 35th Ave | I | 1 | Residential |
| 66 | Osborn/35th Ave | 35th Ave | SB | 2.7 | Osborn Rd | I-10E on-ramp | Fwy | | Mixed |
| 67 | 35th Ave/I-10 East | I-10 East | EB | 2.7 | 35th Ave | I-17 E interchange | Fwy | | Mixed |
| 68 | I-10 East/I-17 East | I-17 East | EB | 3.7 | I-10 East | 19th Ave | Fwy | | Industrial |
| 69 | I-17 East/19th Ave | 19th Ave | NB | 2.2 | I-17 East | Van Buren St | III | 2 | Industrial |
| 70 | 19th Ave/Van Buren | Van Buren St | EB | 2.2 | 19th Ave | 1st Ave | II | 2 | Commercial |
| | End: Van Buren/1st Ave | Total Length | | 180 | | | | | |

Global Positioning System (GPS): Garmin (Kansas City, MO, USA) Map76 GPS to determine vehicle speed and location.

Data Collection System: The laptop computer was used to collect GPS and DustTrak™ data at 1 s intervals in addition to controlling the inlet vacuum pumps.

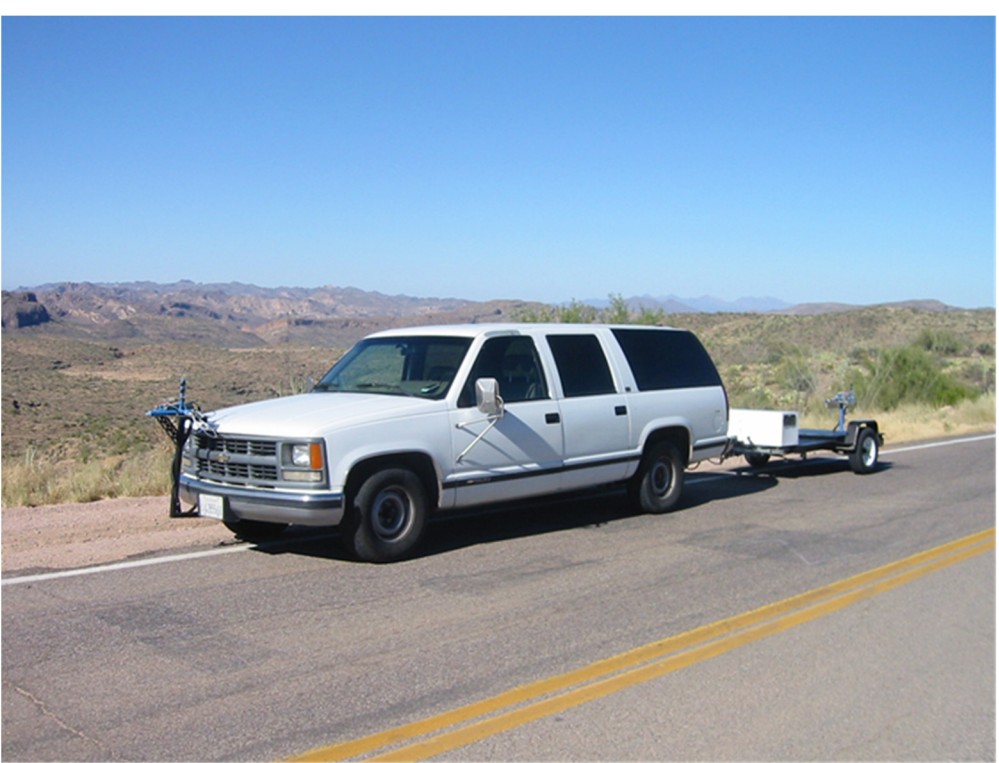

**Figure 2.** Photograph of the SCAMPER.

### 2.3. Data Quality Control and Quality Assurance

The data acquisition system recorded all data accurately. Data were downloaded from the laptop computer and entered into Excel worksheets where all of the calculations were made. Quality control data such as inlet pressure and various voltages were also entered into the master worksheet in addition to GPS location, time, speed, and DustTrak values. Data were validated from logbook entries, and by observing time series, to determine if the results made physical sense. The flow rate and zero of the DustTraks were determined before, after, and during test runs. The drift during the course of each test day was less than a few thousandths of a mg m$^{-3}$, near the 0.001 mg m$^{-3}$ detection limit of the instrument. The instrument is temperature sensitive and therefore the zero drift may be different for moving and stationary modes. The data for each test run were corrected for zero offset using the mean zero response for that day. Two DustTraks were operated collocated at the rear during one test day to determine the precision of these instruments.

There were occasional periods when the GPS did not report data, most likely due to interferences in the sight path to a satellite. In these cases, the cell was filled with the average of the position before and the position after. The same was done for speed and PM. We found that the output of the rear DustTrak occasionally spiked, either positive or negative, most likely due to physical shock. These spikes always showed up for two consecutive seconds. These were unlikely to be associated with an actual PM concentration as concentrations rarely change to that degree in less than one second. This two-second characteristic of this noise spike is also expected from the internal averaging and output characteristics of the DustTrak. On the time constant we selected (which is the shortest available) the DustTrak output is a two-second running average that is updated every second. A large spike in a one-second period will therefore show up as two smaller spikes for two consecutive seconds. To filter this noise, we tabulated the data as five-second running medians. Two-second anomalous spikes therefore would be removed from the data set.

Figure 3 is a plot of the emission rates determined by operating two DustTraks collocated at the rear sampling position for one test day, June 19th. The values from the DustTraks are well-correlated with a slope near unity and an $R^2$ value of 0.96. Other days produced similar results.

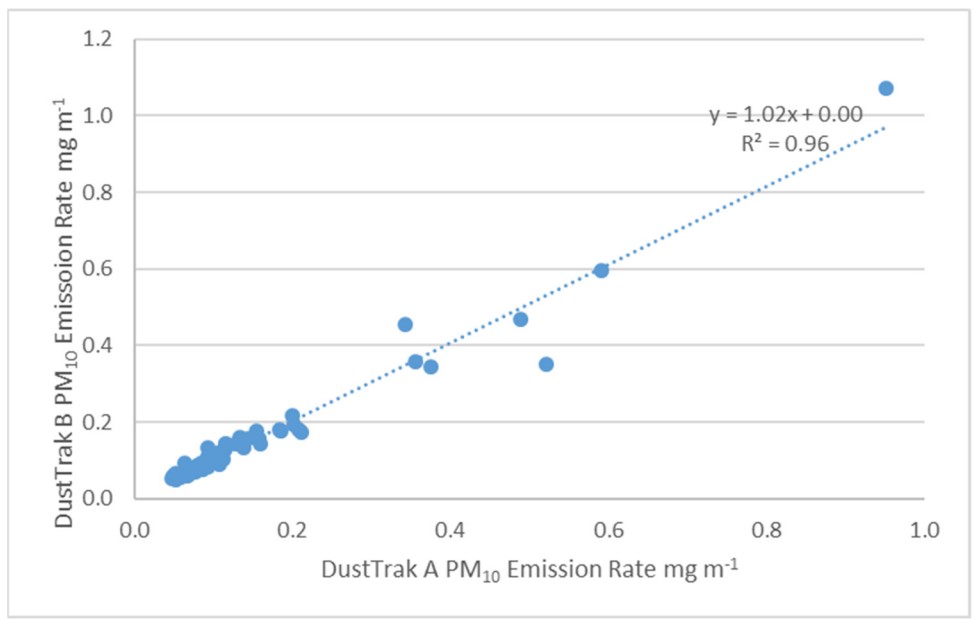

**Figure 3.** Scatter plot of $PM_{10}$ emission rates determined from collocated rear DustTraks.

The differences between the front and rear DustTraks were calculated and the results were multiplied by the frontal area of the Suburban ($3.66 \text{ m}^2$), to yield the emission rate in $\text{mg m}^{-1}$. The PM emission rates for speeds less than $16 \text{ km hr}^{-1}$ were excluded from further analysis since they would be considered unreliable as the production of a well-mixed and defined plume behind the test vehicle was unlikely. This speed was determined visually by watching the test vehicle driven on an unpaved road as various speeds. The emission rate data were then sorted by segments of the route based on the GPS location at the time of data recording. Summary worksheets were prepared that included only time, location, speed, and $PM_{10}$ emission rates.

During three of the test weeks, a short loop was run repeatedly when encountered during the test day. The 8 km long precision test loop was located at the southeast corner of the test route and consisted of segments 18–21. It was chosen to give relatively high emission rates due to the nearby construction activities. The precision of the measurement was determined from these test loops. Precision can also be determined by evaluating the day-to-day variability in the segment and loop-averaged PM emission rates. Since the PM-producing potential of the segments may vary daily due to activities, this evaluation may not fully represent measurement variability.

The response of the DustTraks were calibrated at the factory using Standard Reference Material 8632 from the U.S. National Institute of Standards and Technology. The mass-specific light-scattering response drops rapidly with increasing particle size for particles larger than 1 μm diameter, thus a small change in the particle-size distribution can change the response significantly. Since most $PM_{10}$ regulations are based on collected mass, it was useful to relate the DustTrak output to a mass-based emission rate. A filter-based $PM_{10}$ sampler was therefore operated collocated with the DustTrak mounted on the trailer. $PM_{10}$ filter samples (47 mm Teflo™ Ringed Filter, 2 μm pore) were collected using a Sierra Andersen model 241 inlet adapted to a 47 mm filter holder and sampled at $16.7 \text{ l m}^{-1}$. Filters were equilibrated to 25 °C and 40% RH and weighed before and after collection to the nearest microgram using a Cahn (Irvine, CA, USA) model C-25 electro-balance. Filters were changed based on visual examination to ensure that sufficient material had been collected

to allow for accurate mass determination and to facilitate a broad range of concentrations so that a linear correlation would be meaningful. The average $PM_{10}$ concentration was determined from the DustTrak response during the entire sampling period.

## 3. Results

### 3.1. Precision Test Loop

Table 2 summarizes the results from 90 circuits of the test loops. While the mean $PM_{10}$ emission rates were quite high in March, they dropped progressively during the year as construction activities changed. It should be noted that the relative standard deviation for one of the test days in March was four times higher than any of the other four days. Removing this single day results in a relative standard deviation of 18% for the March tests. These precision results are typical of those determined from the entire route as described in the following section.

**Table 2.** Summary of $PM_{10}$ emission rates results from the precision test loop.

| Date | # Circuits | Mean Emission Rate mg m$^{-1}$ | Mean Standard Deviation mg m$^{-1}$ | Relative Standard Deviation % |
|---|---|---|---|---|
| March | 33 | 1.02 | 0.32 | 38 |
| September | 38 | 0.111 | 0.029 | 23 |
| December | 19 | 0.032 | 0.013 | 41 |

### 3.2. Summary of Emission Rate Data

The test route was traveled once per day, typically starting at approximately 8 a.m. Figures 4–7 are plots of the mean $PM_{10}$ emission rates by segment for all of the test runs. Some segments are missing due to construction activities and detours that caused speed to be generally below 16 km h$^{-1}$. For the March testing, the overall $PM_{10}$ emission rate was 0.094 mg m$^{-1}$ with a relative standard deviation of 21%; these values are similar for the other three test periods.

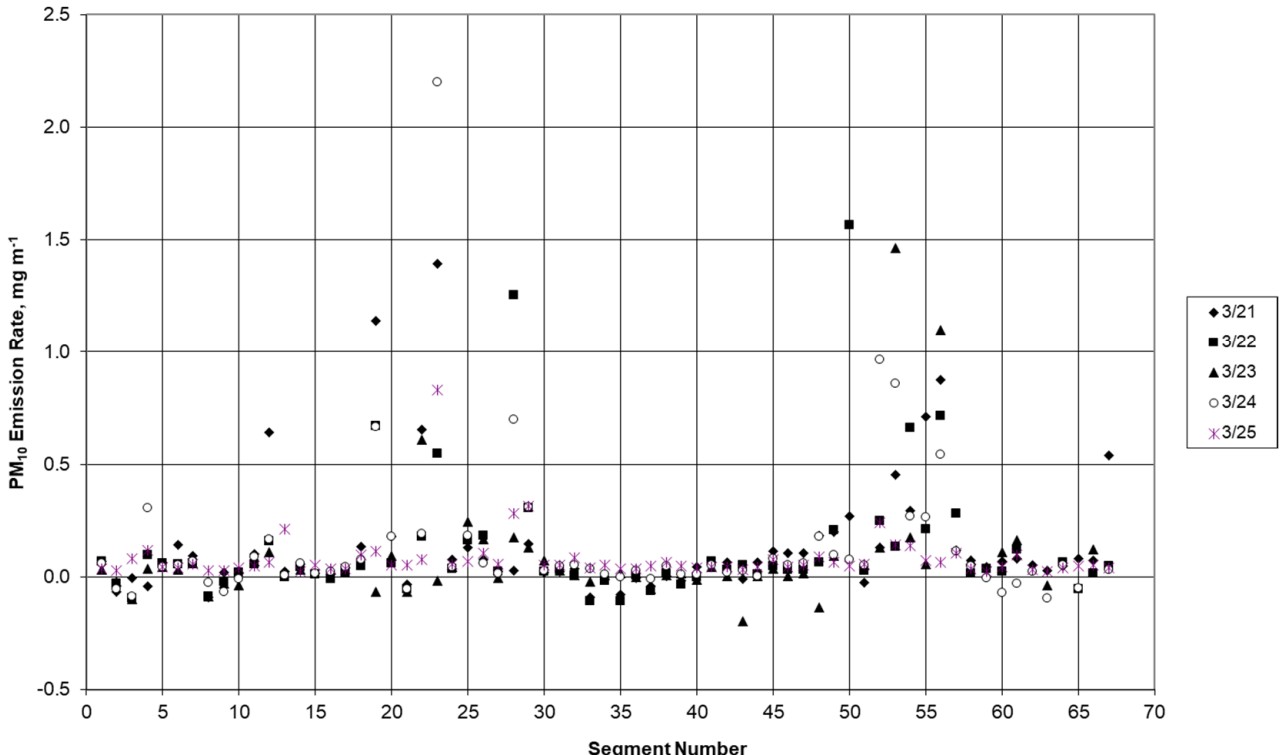

**Figure 4.** Plot of $PM_{10}$ emission rate by segment number starting Tuesday, 21 March.

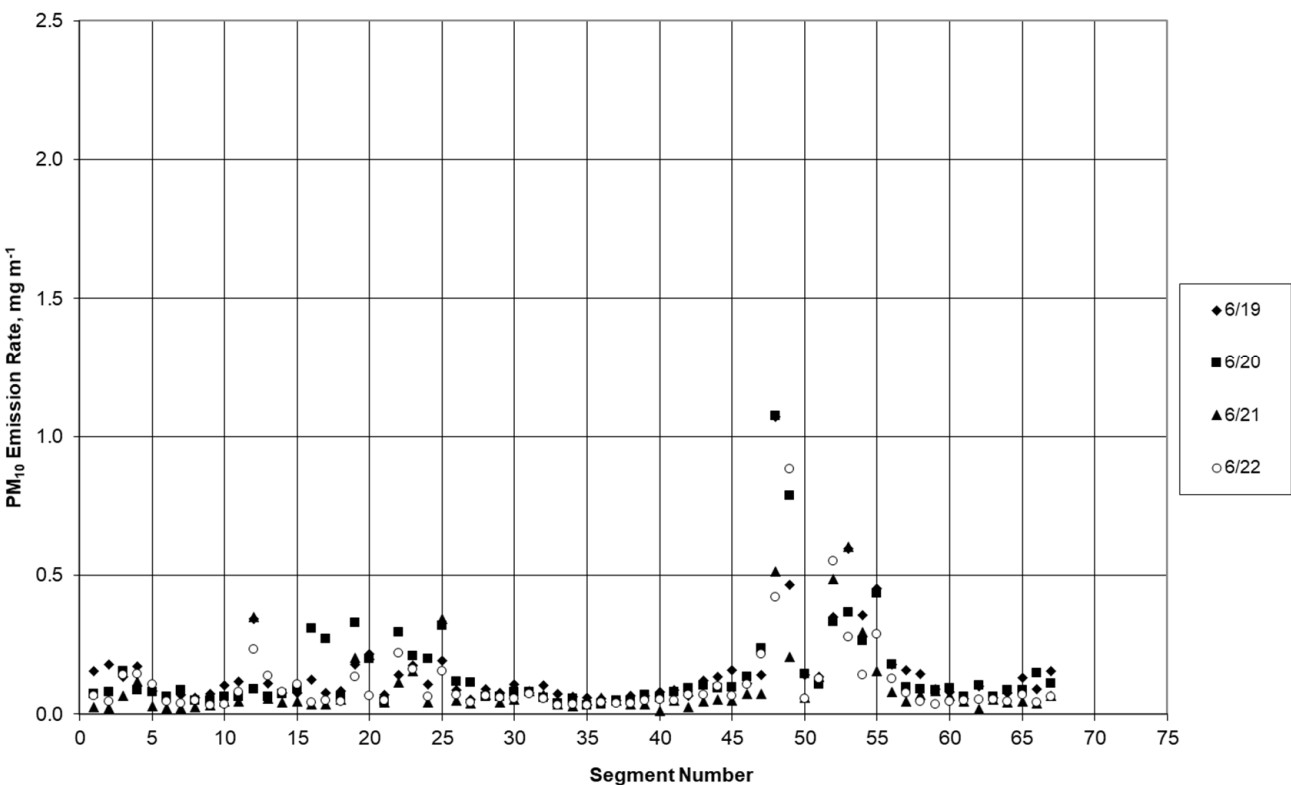

**Figure 5.** Plot of PM$_{10}$ emission rate by segment number starting Monday, 19 June.

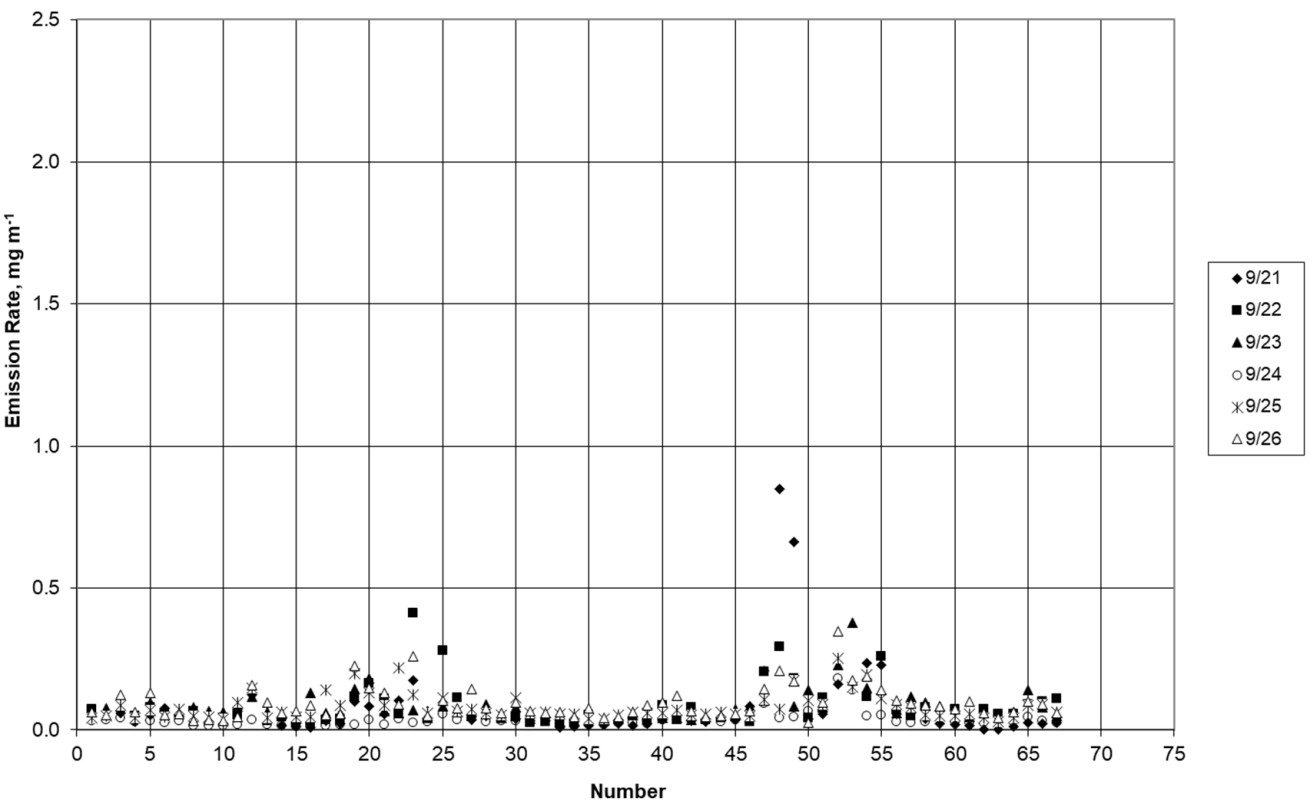

**Figure 6.** Plot of PM$_{10}$ emission rate by segment number starting Thursday, 21 September.

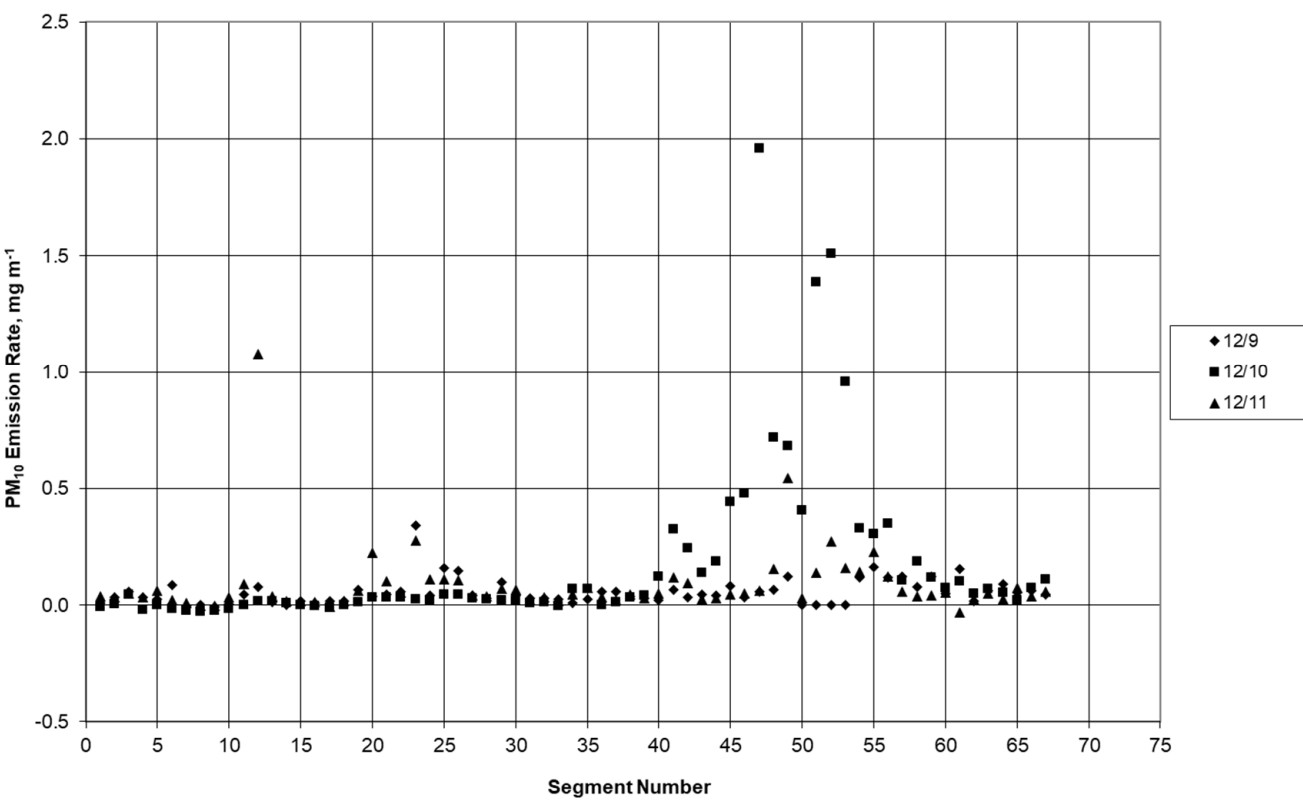

**Figure 7.** Plot of $PM_{10}$ emission rate by segment number starting Saturday, 9 December.

It is clear that most of the $PM_{10}$ emissions are due to a relatively small number of segments, and that these high values are generally repeated for the same segments and groups of segments for each day and for each season. The segments with high $PM_{10}$ emissions were also highly variable in magnitude. This would be expected because of the sporadic nature of activities that deposit soil onto roadways. Note that during the first week of testing there were a number of negative emission rates. This was most likely due to SCAMPER following earth moving or other heavy duty diesel vehicles with noticeable exhaust smoke. These vehicles were avoided in the three remaining seasonal test sessions.

The drop in $PM_{10}$ emission rates on Saturday 25 March is quite noticeable, with only segment #23 rising significantly above the other values. This may be due to a lowered amount of dust generation or deposition from construction activities on weekends. Excluding this anomalous day, the mean $PM_{10}$ emission rate rises to 0.10 with a relative standard deviation of 16%.

In order to more fully examine this potential weekend effect, two study periods in September and December included weekends. Figure 4 shows a tendency for Saturday and Sunday (23 and 24 September) to have lower emission rates. Figure 8, shows a time-series plot of all valid data (not averaged by segment) for each day from Friday through Monday. It is clear that the $PM_{10}$ emission rates drop from Friday to Saturday and drop further from Saturday to Sunday. By Monday the emission rates rise to nearly that of Friday and typical for weekdays. This weekday–weekend effect is very significant as it shows that the $PM_{10}$-producing potential of the roadway can change rapidly.

During the tests conducted in December, the Sunday test was compromised by high winds during the later portions of the test route, with gusts over 40 km h$^{-1}$ which caused dust to be blown over the roadway. The high emission rates observed in Figure 5 are therefore not consistent with other test days.

The mean segment-averaged $PM_{10}$ emission rates were sorted by the five roadway classes based on the ADT by test period. The results are shown in Table 3 along with the overall mean when all test periods were combined and when all classes were combined for a test period. The standard deviations are also included. As indicated by the standard

deviations, there was an expected large amount of variability in emission rates. The class IV roadways (≥30,000 ADT) had the lowest emission rates followed by the freeways. The emission rate for the other classes went up as the ADT lowered. There was no significant variability between the seasons, although the December measurements were biased high due to high winds causing blowing dust in the later segments.

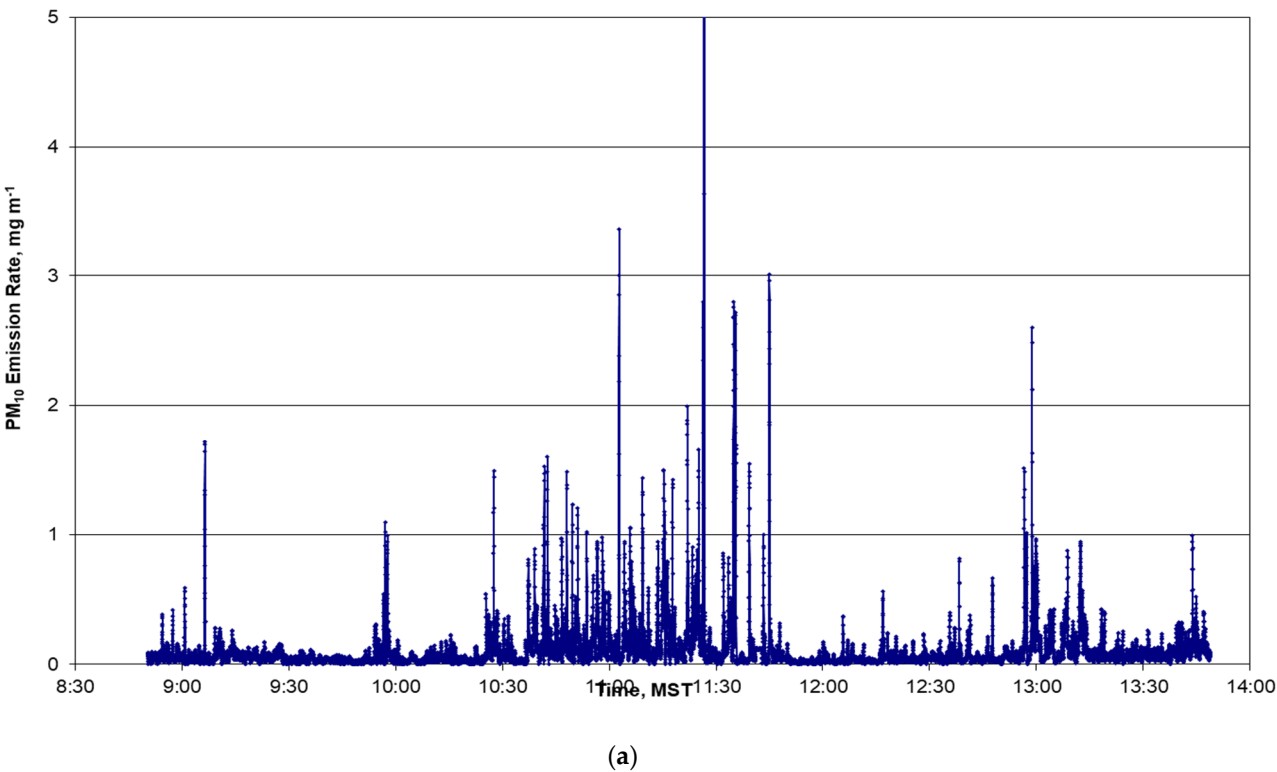

(**a**)

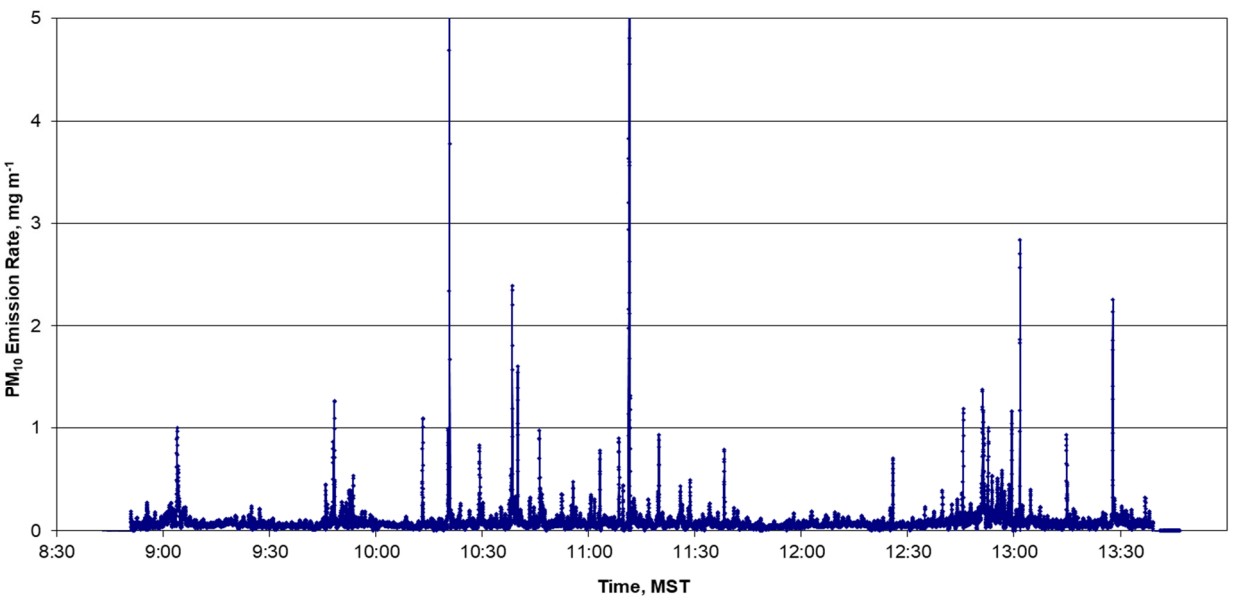

(**b**)

**Figure 8.** *Cont.*

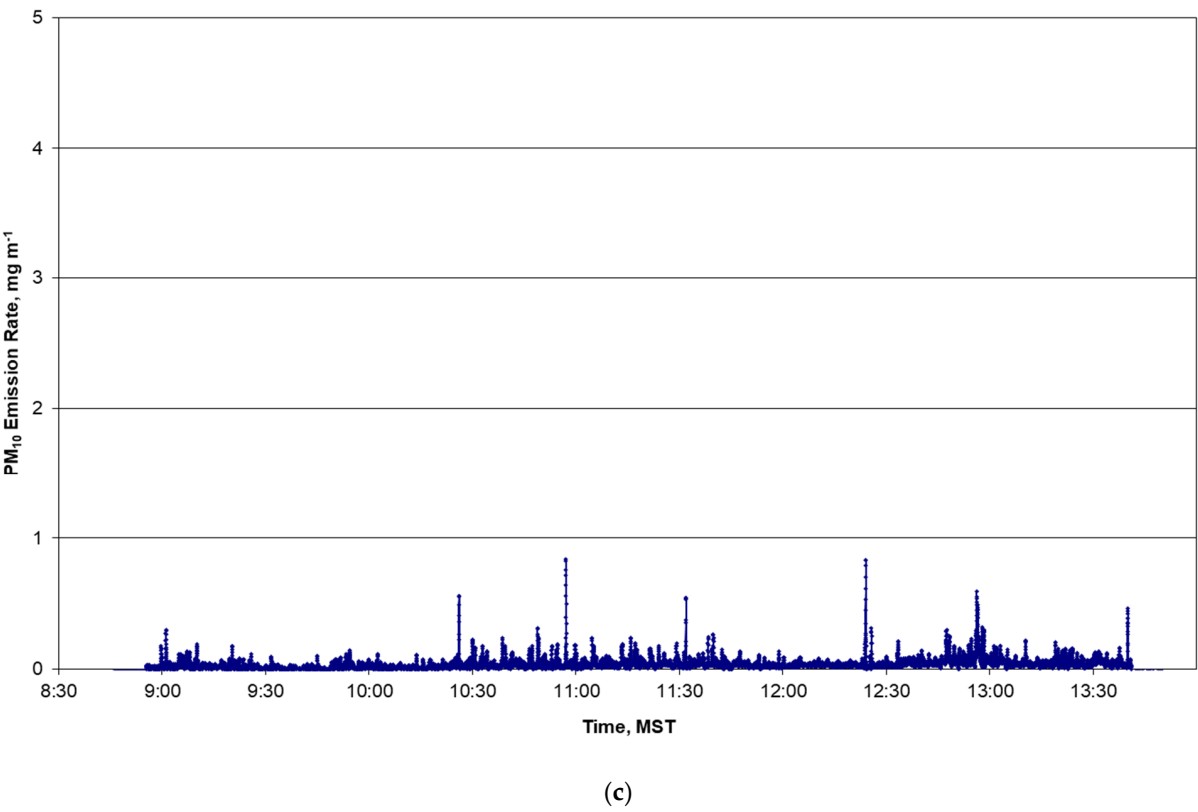

(**c**)

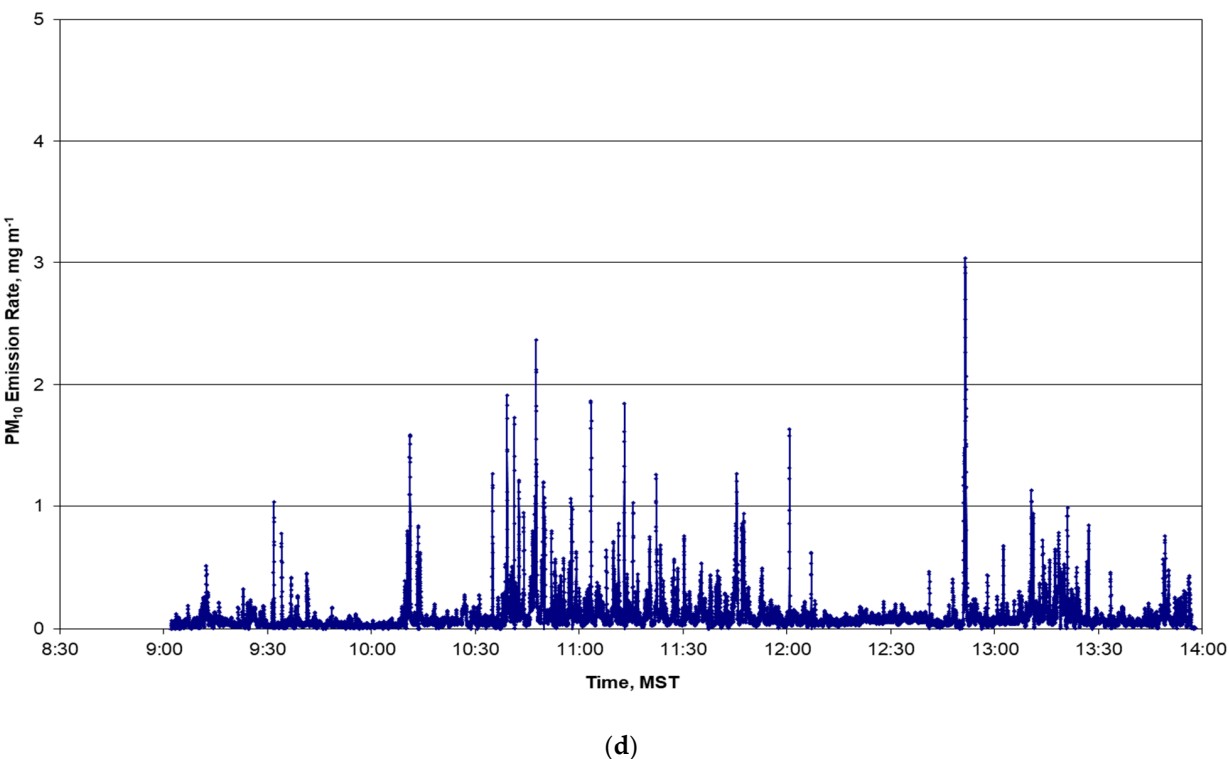

(**d**)

**Figure 8.** Time series plots of $PM_{10}$ emission rates for Friday, 22 September (**a**), Saturday, 23 September (**b**) Sunday, 24 September (**c**), and Monday, 25 September (**d**).

**Table 3.** Summary of $PM_{10}$ emission rates by season, all four seasons, and all five road types combined.

| Road Type | Measurement | March | June | September | December | Combined |
|---|---|---|---|---|---|---|
| Freeway | Mean, mg m$^{-1}$ | 0.03 | 0.07 | 0.05 | 0.03 | 0.05 |
| Freeway | Std Dev, mg m$^{-1}$ | 0.06 | 0.03 | 0.03 | 0.03 | 0.04 |
| $\geq$30,000 ADT | Mean, mg m$^{-1}$ | 0.00 | 0.05 | 0.04 | 0.00 | 0.02 |
| $\geq$30,000 ADT | Std Dev, mg m$^{-1}$ | 0.01 | 0.03 | 0.01 | 0.02 | 0.02 |
| 20,000-29,999 ADT | Mean, mg m$^{-1}$ | 0.06 | 0.08 | 0.05 | 0.08 | 0.07 |
| 20,000-29,999 ADT | Std Dev, mg m$^{-1}$ | 0.06 | 0.04 | 0.02 | 0.09 | 0.05 |
| 10,000-19,999 ADT | Mean, mg m$^{-1}$ | 0.13 | 0.12 | 0.07 | 0.08 | 0.10 |
| 10,000-19,999 ADT | Std Dev, mg m$^{-1}$ | 0.17 | 0.10 | 0.05 | 0.15 | 0.12 |
| <10,000 | Mean, mg m$^{-1}$ | 0.17 | 0.18 | 0.10 | 0.19 | 0.16 |
| <10,000 | Std Dev, mg m$^{-1}$ | 0.24 | 0.20 | 0.10 | 0.35 | 0.22 |
| All Five Combined | Mean, mg m$^{-1}$ | 0.09 | 0.13 | 0.07 | 0.11 | |
| All Five Combined | Std Dev, mg m$^{-1}$ | 0.02 | 0.05 | 0.02 | | |

### 3.3. Comparison of DustTrak $PM_{10}$ with Filter Samples

A large amount of scatter was observed when plotting the DustTrak measurements with filter-based ones. This is not unexpected since the relationship between the two is not linear with a changing particle-size distribution of the various $PM_{10}$ emission sources encountered on the roadway. In addition, the cut-point of the filter sampler may vary with vehicle speed since the size-selective inlet was not designed for the range of speeds encountered on the test route. For this reason, all 46 pairs generated during the four seasonal test periods were plotted as shown in Figure 9. In the figure the least squares correlation line is forced through the origin since filter data precision would be significantly poorer with little collected material. There is considerable scatter, as expected, with an $R^2$ value of 0.46 showing a weak correlation. The slope indicates that the DustTrack data would need to be multiplied by 3.6 to be related to mass emission rates. This is consistent with a factor of 3.5 derived by comparing the mean concentrations.

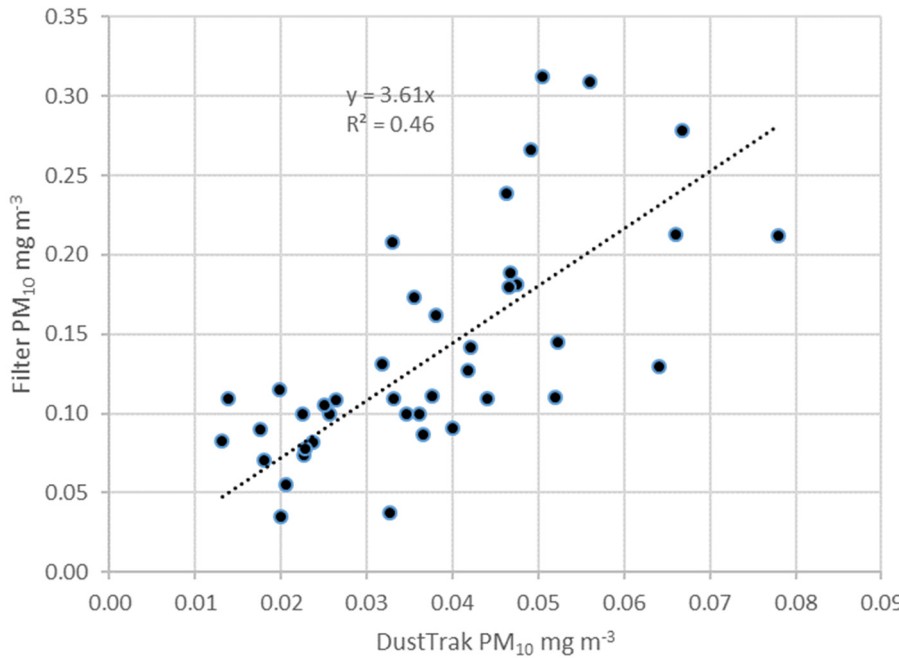

**Figure 9.** Comparison of filter-based and integrated DustTrak $PM_{10}$ concentrations for all test periods.

## 4. Discussion and Conclusions

The SCAMPER mobile-based PM emission measurement approach has been used to fully characterize $PM_{10}$ emissions from paved roads in Phoenix, AZ metropolitan area. $PM_{10}$ emission rate measurements were made at 5-s resolution for over 3200 km of roads with a precision of approximately 25%. It would not be economically feasible to develop such a robust data set by performing silt measurements on the roads. The $PM_{10}$ emission rates varied by over two orders of magnitude and were generally low unless the road was impacted with dust deposited by activities such as construction, sand and gravel operations, agriculture, and vehicles traveling on or near unpaved surfaces (e.g., dirt parking areas and shoulders). These impacted roads were clearly indicated in the data throughout the study, but would be difficult to determine without a mobile measurement system. It is unlikely that significant emissions are due to brake or tire wear from the tow vehicle since there were many periods where the measured emissions were near the detection limit despite high speeds on limited-access roads and braking maneuvers on the other roads. In addition, previous testing on damp roads showed no measurable PM0 emissions. There was no indication that the $PM_{10}$ emission rate varied significantly with season. This is not unexpected since the Maricopa County climate leads to very dry conditions most of the time.

Of particular significance was the much lower $PM_{10}$ emission rate on a weekend and especially Sunday, when dust generating activities such as construction are much reduced. This indicates that roads reach an equilibrium $PM_{10}$ emission potential within a day. A rapid decline, within a few vehicle passes, of $PM_{10}$ emissions have been reported using porous silica particles of various sizes impregnated with a fluorescent dye tracer [24]. By measuring the fluorescent intensity after each vehicle passes, it was reported that over half the particles greater than 10 μm were removed after the first two vehicles passed at 64 km $h^{-1}$. Further passes had little effect on the amount of deposit. Using mobile methods on a roadway with deposited soil, Fitz et al. [23] reported a similar rapid decrease after several vehicle passes followed by a slow decline in $PM_{10}$ emission rates.

The sporadic nature of emissions near construction activities and the fairly rapid decrease when these activities lessen need to be taken into account when developing emission inventories and mitigation methods. The SCAMPER could be useful in identifying high emission episodes in real-time and verifying that the applied mitigation methods are effective.

The DustTrak most likely measures lower $PM_{10}$ concentrations compared with weighed filter samples under these conditions because the larger particles found in suspended road dust scatter light much less efficiently than the particles used to calibrate the DustTraks. In our previous study it was found that the correction factor for the DustTrak was 2.8 when sampling next to a roadway with artificially-deposited material and 2.4 when the same material was re-suspended in a laboratory [23]. In another study using the SCAMPER in Las Vegas, NV on public streets the factor was found to be 1.25 [22]. All of these tests showed considerable scatter. If the SCAMPER results need to be related to regulatory-defined $PM_{10}$ mass concentrations, then collocated filter sampling is recommended.

The EPA has a list of default silt loading values for normal baseline conditions which are widely used in emission inventories. Using equation 1 and assuming that $W^{1.02}$ is 2 results in the following emission rates in units of mg $m^{-1}$: <500: 0.78; 500–5000: 0.29; 5000–10,000: 0.095; >10,000: 0.051; limited access: 0.027. Since the ADT breakdowns are not consistent with the ones that we used, it is not possible to do a direct comparison. For the limited access freeway, the mean SCAMPER value was 0.05 mg $m^{-1}$. Applying the factor of 3.5 results in an emission rate of 0.18 mg $m^{-1}$, which is nearly 7 times higher than the EPA default value. A factor of 4 results if a Class II roadway is compared to the EPA's > 10,000. While these values are within an order of magnitude, one must consider that the route was chosen to include segments with significant construction activities and therefore high emission potential. In addition, the Phoenix metropolitan area, with an annual growth rate of approximately 3%, is one of the fastest growing in the United States

and therefore has significantly higher construction activities than would be expected of a typical U.S. metropolitan area, for which the default values were presumably intended.

We conclude that the SCAMPER approach can easily and safely generate more appropriate $PM_{10}$ emission rates from paved roads that are specific to a geographical area. The SCAMPER vehicle is representative of the vehicle mix in an urban area and can be driven at a speed typical of the traffic flow. The speed is important since the $PM_{10}$ emission rate has been shown to be highly dependent on vehicle speed [23]. There is no need to barricade lanes and apply labor-intensive and sometimes dangerous silt loading measurements. Emission measurements can also be made on high-speed, limited access roadways for which silt loading measurements are simply not feasible. One of the limitations is that periods of high winds should be avoided since the SCAMPER has not been evaluated under these conditions and erratic results may be obtained that are not representative of normal weather conditions for a given location. Another limitation is that with wet pavement we have observed emission rates that were at or below the detection limit.

The real-time SCAMPER data could be used to improve the accuracy of $PM_{10}$ emission inventories. SCAMPER is a relatively low-cost device to build and operate compared to other measurement approaches. A major advantage is that it has been shown to not require calibration by flux methods [23], which are labor- and equipment-intensive. In addition, activities that may cause high emissions of $PM_{10}$ from roadways can be easily monitored for mitigation and enforcement purposes.

**Author Contributions:** D.R.F. was responsible for funding acquisition, project administration, conceptualization, methodology, formal analysis, and all writing. K.B. performed software integration, data collection, and data validation. All authors have read and agreed to the published version of the manuscript.

**Funding:** Funding was provided by Maricopa County Association of Governments contract # 284.

**Institutional Review Board Statement:** No human or animal studies were involved.

**Informed Consent Statement:** No humans were involved.

**Data Availability Statement:** The finalized data are contained within the article. The raw data do not reside in a publicly available data set, but are available from the corresponding author.

**Acknowledgments:** The authors thank the Maricopa County Association of Governments for funding this project and Cathy J. Arthur for providing the test route and many helpful suggestions.

**Conflicts of Interest:** The authors declare no conflict of interest.

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
