# Peer review of "Characterization of PM10 Emission Rates from Roadways in a Metropolitan Area Using the SCAMPER Mobile Monitoring Approach"

_atmosphere, doi:10.3390/atmos12101332_

Round 1
Reviewer 1 Report
In all studies focused on air quality one of the most uncertain parameters is emission data. Therefore, the attempts to improve our understanding of emissions to air either from industrial sources or from natural processes are highly important. In particular, in the presented research the authors tried to estimate emission flux of dust from paved roads caused by moving traffic. Therefore, the work is interesting from scientific point of view and useful from viewpoint of possible further application of the results in model assessment of emissions form roads and modeling of air quality. I believe that the work can be published. However, I would like to mention some comments that require clarification.
- The purpose of the work is not clearly formulated. For me it remained unclear what was the objective (or objectives) of the work.
- Lines 35-37: the authors mention that PM10 in air can be significantly contributed by material originated from paved roads and support this statement by only one reference related to 1992. I think, there should be other more contemporary works which could be referred.
- All abbreviations must me described. For example, QC (line 158 and further).
- In Section 2.1 some classification of roads according to average daily traffic is presented. However, I do not understand what freeway means. How can I distinguish freeway from other types of roads?
- Another question about ADT. What are units for ADT? Mean number of vehicles for a day?
- Line 152: “The PM emission rates for speeds less than 16 km hr-1 were excluded … “. Why exactly 16 km/h? Some explanation or reference is needed.
- Table 2. All symbols (ER, SD) should be explained. I can guess that SD stands for standard deviation, but what is ER?
- Table 2. Does “combined” is “mean over all periods of tests”? It is good to indicate it clearly in caption to the table.
- Line 165. “The response of the DustTraks was calibrated using Arizona Road Dust”. What is Arizona Road Dust? I think it should be clarified in the text.
- Lines 95-97: “The SCAMPER determines PM emission … using optical sensors with a 1s time resolution”. Line 269: “PM10 emission rate measurements were made at 5-second resolution”. So what time resolution was used – 1 s or 5 s? Probably it is not principle in this research, but I think it would be good to have the consistency in the text.
Reviewer 2 Report
- It would be better for readers to understand the article if some terms and details of equipment are provided in the manuscript even it was already explained in previous studies. For example, full name of SCAMPER and drawing or picture.
- 2.SAMPER description: misspell
- There are several variables to affect the observation in this study. Month, time are considered, but other weather information such as rain, wind speed and direction, temperature, which might influence the measurement, performed under outdoor conditions, were not properly described in the text, or graph. I would like to suggest to discuss other variables at least in analyzing the results.
- Details of filter, target of interest is only PM10, So. Details of filter in terms of the collection amount, sieve size and material
- Component of collected PM10. Even considered the aim of this study focuses on the emission rate, but variables such as construction activity can be regarded as a dominant factor when its amount and compositions are confirmed to be mainly soil dust. Other possible particulate matter sources such as brake pads or disc and tire wear
- Consider the generation, resuspension, and transport of PM in the target site and route in discussing the results of measurement.
- Graphs in figure 6 appears to be dragged horizontally and were distorted with different ratio.
- Explain about the limitations of SCAMPER to overcome for higher reliable data generation
- Edit figures and tables with high resolution image and editable tables, not just copying or capturing of data in other formats. (figure 1, table 1)
